# Family Resilience and Dyadic Coping during the Outbreak of the COVID-19 Pandemic in Italy: Their Protective Role in Hedonic and Eudaimonic Well-Being

**DOI:** 10.3390/ijerph20186719

**Published:** 2023-09-06

**Authors:** Francesca Giorgia Paleari, Irem Ertan, Lucrezia Cavagnis, Silvia Donato

**Affiliations:** 1Department of Human and Social Sciences, University of Bergamo, 24129 Bergamo, Italy; iremertan92@gmail.com (I.E.); lucrezia.cavagnis@unibg.it (L.C.); 2Department of Psychology, Università Cattolica del Sacro Cuore, 20123 Milan, Italy

**Keywords:** family resilience, hedonic well-being, eudaimonic well-being, dyadic coping, COVID-19 pandemic

## Abstract

The COVID-19 pandemic outbreak has dramatically worsened people’s psychological well-being. Our aim was to examine for the first time the concurrent and longitudinal relations of family resilience with hedonic and eudaimonic well-being, and the moderating role of socio-demographics. For people having a romantic partner, we also explored whether family resilience and dyadic coping were uniquely related to well-being. One cross-sectional study (*N* = 325) and one 10-week follow-up study (*N* = 112) were carried out during the outbreak of the COVID-19 pandemic (April–May 2020) in Northern Italy. Adult participants completed an online questionnaire in both studies. Correlation, multivariate regression, and moderation analyses were carried out with IBM SPSS version 28 and its PROCESS macro. Significance of differences in correlation and regression coefficients was tested through Steiger’s procedure, Wald test, and SUEST method. Family resilience was found to relate more strongly to eudaimonic (versus hedonic) well-being concurrently and to hedonic (versus eudaimonic) well-being longitudinally. The concurrent or longitudinal relations with hedonic well-being were generally stronger for females, part-time workers, and people undergoing multiple stressors. For people having a romantic partner, family resilience was concurrently associated with well-being independently of dyadic coping, whereas dyadic coping was longitudinally related to well-being independently of family resilience. Family resilience was found to protect, in the short term, the psychological well-being of people facing the pandemic outbreak. Its protective role mainly concerned hedonic well-being and was more pronounced for more vulnerable people. For persons having a romantic partner, however, dyadic coping seemed to have equal, if not greater, positive short-term effects.

## 1. Introduction

The COVID-19 pandemic evolved into one of the largest public health crises of our time, causing enduring physical and psychological complaints in an incredibly large number of people. The COVID-19 pandemic has indeed dramatically worsened people’s psychological well-being all over the world, both in its hedonic component (i.e., experiencing positive affect and being satisfied with one’s life) and eudaimonic component (i.e., having meaningful purpose in one’s life and a positive orientation toward self-actualization). Through one cross-sectional and one follow-up study, the present research examined whether family resilience (i.e., the ability of the family to withstand and rebound from adversities) and dyadic coping (i.e., the way couples cope with stress together) protected people from a decline in hedonic and eudaimonic well-being resulting from their exposure to the pandemic outbreak in Italy.

### 1.1. Psychological Well-Being Outcomes of COVID-19 Pandemic

The COVID-19 pandemic has had and is currently having a significant impact on public health by impairing people’s psychological well-being [1]. Numerous studies conducted across the world have documented higher levels of depression, negative affect, anxiety, stress, and mood disorders, as well as lower levels of subjective well-being in the general population during the first pandemic waves, compared to the pre-pandemic period (e.g., [2,3,4,5,6]). These negative outcomes pertain to the hedonic component of well-being [7], conceptualized as experiencing positive affect instead of negative affect [8] and being satisfied with one’s life [9]. Psychological well-being, however, is not simply the presence of subjectively positive cognitive–affective experiences, rather it is the “combination of feeling good and functioning effectively” ([10], p. 137). The concept of functioning effectively reflects the eudaimonic component of well-being, that is, a positive orientation and functioning which leads to success in the face of life’s existential challenges [7]. Eudaimonic well-being is characterized by a sense of continued growth, development, and flourishing as a person, by the belief that one’s life is purposeful and meaningful, by a sense of authenticity and self-determination, by the pursuit of excellence, and by experiencing positive relationships [11,12]. Although less investigated than hedonic well-being, eudaimonic well-being was also observed to decline in the aftermath of the pandemic outbreak; autonomy and positive relations were particularly compromised, which was consistent with the reduced freedom and opportunities for social interactions during lockdowns (e.g., [13,14]). 

The pandemic differentially affected people’s psychological well-being depending on a variety of characteristics such as their gender, age, involvement in a romantic relationship, housing conditions, political orientations, employment status, exposure to work-related risk of infection or to multiple stressors. In particular, anxiety and general mental health worsened significantly more for females than for males, because—compared to males—females had to face greater socioeconomic challenges during the pandemic, being in more precarious economic conditions, being overrepresented in health care jobs at risk of COVID-19 infections, providing most of the child and family care, and being more often victims of increased intimate partner violence [15].

Emerging and young adults experienced a greater increase in negative affect, depression, anxiety, PTSD symptoms and loneliness than older adults, in line with evidence that the latter are generally less affected by adversities due to their propensity to prioritize existing resources (e.g., close relationships) and experiences instead of being involved in the young adult tasks of broadening one’s social network and experiences [16,17,18]. 

Being in an intimate relationship was found to be a protective factor during difficult times. Indeed, married or cohabiting individuals experienced lower levels of anxiety, depression, and stress than single individuals during the COVID-19 pandemic [19,20]. 

Moreover, previous research emphasized the importance that living in homes with adequate spaces [21,22] and with outdoor domestic environments [23,24] during the lockdown had in well-being. Similarly, individuals who were satisfied with the spatial dimensions of their home experienced lower levels of psychological distress [25].

Interestingly, political orientation was also found to be associated with well-being during the COVID-19 pandemic. Indeed, the left-wing political view was associated with stress worsening, whereas the right-wing did not [26]. 

Few studies have also explored the impact of employment status on psychological distress during the COVID-19 pandemic. For example, Walia and colleagues [27] showed that unemployed people reported more psychological distress and alcohol consumption compared to employed ones. Additionally, Mimoun and colleagues [28] argued that people on furlough reported higher distress levels compared to unemployed and full-time workers. Regardless of gender, workers exposed to higher risk of infection showed a worsening of psychological well-being, in terms of more psychological and death distress and less happiness, compared to workers exposed to lower risk of infection [29,30]. 

Finally, there is evidence that undergoing multiple stressors undermines psychological well-being. For example, Rapelli et al. [31] showed that, compared to people without chronic illness during the COVID-19 pandemic, those with a chronic disease had lower psychological well-being, and more fears and worries about COVID-19. 

### 1.2. The Protective Role of Family Resilience

Despite severity, the negative consequences of the COVID-19 pandemic on both hedonic and eudaimonic well-being have been found to progressively reduce over time, probably due not only to actions taken to control the pandemic, but also to adaptation and resilience processes [2]. Indeed, individual resilience, that is the person’s ability to positively cope with adversities, and encourage positive readjustment and development when facing challenging situations [32], has been shown to predict greater hedonic and eudaimonic well-being reported by the general adult population during the pandemic [33,34,35,36,37].

COVID-19 research mostly focuses on resilience conceived and measured as a person’s characteristic. However, overcoming an individualistic approach to the conceptualization of resilience and adopting a transactional social–ecological perspective to it, a growing number of scholars claims that the ability to be resilient transcends the individual and emerges from their wider social environment, including the family system (e.g., [38,39,40]). In this regard, Walsh defined family resilience “as the ability of the family, as a functional system, to withstand and rebound from adversity” [41], an active process of transformation by which family members and their relations emerge strengthened and more resourceful through the family members’ shared efforts in confronting challenges [42]. According to Walsh’s theoretical framework [43], three domains of family functioning contribute to family resilience and are key components of it: family belief systems (i.e., making meaning of adversity, positive outlook, and transcendence), family organization (i.e., flexibility, connectedness, and social and economic resources), and family communication (i.e., clarity, open emotional expression, and collaborative problem-solving). 

Emerging evidence indicates that family resilience is positively related to family members’ mental health in a variety of challenging situations, such as having a child with developmental disorders or a female family member with breast cancer [44,45,46,47]. Some scholars have argued that family resilience may similarly facilitate family adjustment to the pandemic challenges [48,49,50], but only a few studies have investigated family resilience components in relation to well-being outcomes during the COVID-19 pandemic. These studies have documented that family resilience and its dimensions (family beliefs systems, organization, communication, and problem-solving) are cross-sectionally associated with mental health issues like negative affect, depression, anxiety, stress symptoms, poor sleep, pandemic burnout, and PTSD among community children, adolescents, adults [51,52,53,54,55,56], and oncology patients [57]. Furthermore, research also showed that family resilience is also associated with COVID-19 perceived psychological impact when considering the interdependence of family dyads from the same household [58]. However, to our knowledge, research has not yet disentangled the direction of the links between family resilience and well-being through longitudinal data nor has it investigated the potential association of family resilience with the eudaimonic components of well-being. Given that the family resilience and eudaimonic well-being constructs share many common features including strengthened relations, life perceived as meaningful, positive transformation, and growth [59], it is possible that family resilience relates to eudaimonic more strongly than to hedonic well-being. In light of the importance of eudaimonic well-being features in face of unprecedent life challenges, understanding factors that can sustain this specific form of well-being is crucial. Also, very few studies have so far explored for which kind of population family resilience might be more effective in protecting from COVID-19 undesirable mental health outcomes. In this regard, Giordano et al. [60] found that family resilience was associated with children’s emotional symptoms reported by parents only if they were married or cohabiting, not if they were divorced. Identifying such boundary conditions would be of primary practical importance, as it would allow identifying the circumstances in which implementing interventions aimed at enhancing family resilience could be more useful.

### 1.3. The Protective Role of Dyadic Coping

Like family resilience, dyadic coping has also been found to relate to lower psychological distress and higher hedonic well-being during the COVID-19 pandemic [61,62,63]. Conceptualized as the way couples cope with stress together by sharing appraisals of demands and strategies to face them [64], dyadic coping is assumed to relate to family resilience but also to differ from it in many respects. Whereas family resilience entails a positive adaption to stressful situations [41], dyadic coping can be either positive (e.g., supportive, common, and delegated) or negative (e.g., hostile, ambivalent, or superficial) [65]. Whereas family resilience typically requires time to emerge and display its effects, because it results from the development of new individual and relational strengths needed to re-establish an equilibrium in the family system subsequent to disturbances caused by adversities [41,66], dyadic coping is a strategy to manage stressors which can be adopted and show its outcomes right after the onset of demanding situations. In this regard, diary studies indicate that dyadic coping is associated with mood and relational outcomes on a daily basis [67,68], even in the immediate outbreak of the COVID-19 pandemic [69]. The rare longitudinal studies conducted on family resilience development, instead, show that processes through which families become more resilient and positively adapt to adversities typically occur over much longer periods of time [70].

Despite family resilience and dyadic coping being supposed to be related, different frameworks provide different views about how this might happen. For example, according to Walsh’s model, positive dyadic coping is one of key communication and problem-solving strategies which contribute to family resilience [41,49], while Afifi and colleague’s theory of resilience and relational load [71] assumes that family resilience can lead to a greater communal orientation and more effective dyadic coping. 

Surprisingly, despite their close interconnections, family resilience and dyadic coping have been rarely investigated simultaneously (for a few exceptions see [72,73]) and, to the best of our knowledge, their unique contribution to psychological well-being has never been examined so far.

### 1.4. Research Aims and Hypotheses

On the basis of this review, our first goal was to test whether family resilience was differently associated with hedonic and eudaimonic well-being both concurrently and longitudinally in the aftermath of the COVID-19 pandemic outbreak in Italy. Given that, compared to hedonic well-being, eudaimonic well-being shares more common features with family resilience, like strengthened relations, life perceived as meaningful, positive transformation, and growth [59], we expected that family resilience related more strongly with eudaimonic than hedonic well-being.

A second aim was to explore whether the above concurrent and longitudinal associations of family resilience with hedonic and eudaimonic well-being were moderated by a wide range of socio-demographic and pandemic-related characteristics. Due to the lack of substantial empirical evidence of the boundary conditions at which family resilience is effective in protecting from COVID-19 undesirable mental health outcomes, we were unable to make specific assumptions on the topic. On the one hand, people whose mental well-being was shown to be most at risk during the pandemic—such as women, youth, singles, unemployed, workers at higher risk of COVID-19 infection, people having a left-wing political orientation, living in poor housing conditions, or undergoing multiple stressors [15,16,17,18,21,22,23,24,25,26,29,30,31,74]—might have benefitted more from a resilient family functioning that provided them relational resources and support to confront with the pandemic. On the other hand, since people at high risk of well-being impairment were typically struggling with multiple stressors during the COVID-19 pandemic compared to people at low risk, having a resilient family environment might not have been enough to protect their well-being, especially if they were lacking in personal coping and resilience skills. In this regard, Zhuo and colleagues [75] showed that family resilience was positively related to adolescents’ mental health during the pandemic only when adolescents’ ability to notice, distinguish, reflect, and control their emotions was high.

A third and last aim was to examine in subjects having a romantic partner whether family resilience and dyadic coping were uniquely associated with hedonic and eudaimonic well-being, one independently of the other. The lack of studies investigating the unique contribution of family resilience and dyadic coping to psychological well-being also prevented us from making specific assumptions for this third aim. On the one hand, one can assume that family resilience is linked to well-being independently of positive dyadic coping, because family resilience is a broader and more comprehensive construct that subsumes many other components—related to the family belief system, organization, communication, and problem-solving strategies [43]—besides dyadic coping. On the other hand, one can also hypothesize that dyadic coping is related to well-being independently of family resilience because, for people involved in a romantic relationship, the couple dynamics have a stronger impact on their personal well-being compared to family functioning. Therefore, positively addressing pandemic adversities with the partner may be more relevant to their psychological well-being than being able to conduct it with the whole family, especially in individualistic societies where the couple relationship is a particularly important source of personal well-being (e.g., [76,77]). 

To address these aims, we conducted two studies in the Northern Italian districts which were the epicenter of the pandemic in Europe. Study 1 was a cross-sectional study in which adult participants reported their socio-demographics and pandemic-related information as well as their levels of hedonic and eudaimonic well-being, family resilience, and, for those with a romantic partner, dyadic coping in the aftermath of the pandemic outbreak (April–May 2020), during the first national lockdown in Italy. Study 2 was a follow-up study in which participants reported their level of hedonic and eudaimonic well-being 10 weeks after Study 1 data collection, when the first national lockdown was over.

## 2. Study 1

Study 1 aimed at investigating cross-sectionally whether: (a) family resilience was differently associated with hedonic and eudaimonic well-being; (b) the associations of family resilience with hedonic and eudaimonic well-being were moderated by socio-demographic and pandemic-related characteristics; and (c) for subjects having a romantic partner, family resilience and dyadic coping were related to well-being one independently of the other.

Relying on the literature previously reviewed, we expected that family resilience was more strongly associated with eudaimonic well-being than with hedonic well-being, since it shares with the former features like strengthened relations, perceived life meaningfulness, positive transformation, and growth [59]. However, due to a lack of consistent evidence, we did not make specific assumptions about which specific socio-demographic and pandemic-related conditions may moderate the relationship between family resilience and hedonic and eudaimonic well-being, nor whether family resilience and dyadic coping were associated with hedonic and eudaimonic well-being one independently of the other.

### 2.1. Materials and Methods

#### 2.1.1. Participants and Procedure

Participants were 325 adults (76.0% females), averaging 29.90 years old (*SD* = 12.8; range 19–73) and living in Northern Italy (98.2%), mostly in the districts which had been severely affected by the first wave of the COVID-19 pandemic in Europe (76.3%). Their socio-demographic and pandemic-related characteristics are reported in Table 1.

Characteristics of participants involved in a romantic relationship (*N* = 212; 65.2%) were similar to those of respondents not having a romantic partner, except for their age (*t*-test (322) = 6.345, *p* = 0.000), employment status (*χ*^2^(3) = 50.10, *p* = 0.000), and their cohabitation with parents (*χ*^2^(1) = 24.90, *p* = 0.000). Participants having a romantic partner were older (*M* = 32.6 vs. *M* = 24.8), and were more likely to have a part-time or a full-time job (65.1% vs. 30.4%), and not to live with their parents (54.7% vs. 83.2%). 

Participants were contacted through an invitation message posted on social networks or sent by messaging apps. The invitation indicated that the purpose of the study was to examine the effects of the COVID-19 outbreak on personal well-being and that participants had to be Italian and at least 18 years old. Participants were then invited to complete a questionnaire, which took approximately 30 min and was implemented online using the Google Forms platform. Data collection was completely anonymized and it was not possible in any way to trace the respondent. No compensation was offered for participation. 

Data were collected in 2 weeks between April and May 2020 during the first national lockdown period, when mandatory strong restrictive measures were applied throughout Italy (e.g., obligation to stay at home, and to study or work remotely). During the data collection period, 21,395 Italians were diagnosed with COVID-19 and 3,916 died after being infected, mostly in the districts where participants in this study lived.

After completing the questionnaire, participants were asked to share their email address if they consented to be contacted for a follow-up data collection. Most subjects (67%) agreed to participate to the follow-up study.

All respondents were treated according to the ethical guidelines established by the WMA—Declaration of Helsinki [78] and the Italian Psychological Association [79]. These guidelines include obtaining informed consent from participants, maintaining ethical treatment and respect for their rights, and ensuring the privacy of participants and their data.

#### 2.1.2. Measures

*Sociodemographic and pandemic related characteristics.* Participants were asked to report: their age, sex, degree, employment status, if they had lost their job due to the pandemic, (if working) the risk level of contracting COVID-19 by doing their job, if they had a (cohabiting) romantic partner, their marital status, if they lived with their parents, if they lived with different people than usual due to the pandemic, the size of their house and garden, their political orientation, and if they had experienced distressing events not caused by the pandemic since its outbreak in Italy, that is, approximately in the last two months (for details regarding response options see Table 1).

*Family resilience*. We used the 26-item Italian Version of the Walsh Family Resilience Questionnaire [80] to assess three key dimensions of family resilience: shared belief and support (item example: “We try to make sense of our crisis situation and our choices”, “We can count on the fact that family members will help one another in difficulty”), family organization and interaction (“We are flexible in facing unforeseen events and adapting to new challenges”, “We can show understanding, accept differences, and avoid negative judgements”), and utilization of social resources (“We can trust in the help of relatives, friends, neighbors, and the community”, “We have economic security to be able to overcome difficult times”). Respondents were asked to rate the extent to which each item described their family on a 5-point Likert-type scale ranging from 1 (*very little*) to 5 (*very much*). Since the scores on three dimensions were strongly correlated (0.62 ≤ *r*s ≤ 0.89), mirroring the results found in the Italian validation study [80], they were averaged in one overall index (α = 0.96). Higher scores indexed higher family resilience.

*Dyadic coping.* To assess dyadic coping, the Italian version of the Dyadic Coping Inventory [81,82] was used. This 37-item questionnaire measures perceived communication and dyadic coping (supportive, delegated, negative, and common) that occur in dyads when one or both partners are distressed. Specifically, the scale consist of 14 items assessing dyadic coping provided by oneself (e.g., “When my partner is feeling stressed, I show empathy and understanding to him/her”), 14 items assessing dyadic coping provided by the partner (e.g., “When I feel stressed, my partner shows empathy and understanding me”), 5 items assessing joint dyadic coping (e.g., “When we both feel stressed, we try to cope with the problem together and search for ascertained solutions”), and 2 items evaluating the quality of self-perceived dyadic coping (e.g., “I am satisfied with the support I receive from my partner and the way we deal with stress together”). The items were administered on a 5-point scale (from 1 = *never* to 5 = *very often*). A total score was obtained by averaging the 35 items (excluding the 2 evaluation items), after reverse coding items assessing negative dyadic coping (α = 0.93). As a result, higher scores represented higher positive dyadic coping. 

*Psychological well-being*. The Italian version of the Psychological General Well-Being Index (PGWBI) and the Psychological Well-Being (PWB) scales were used to assess hedonic and eudaimonic psychological well-being, respectively [83]. 

The PGWBI [84], cross-culturally validated for use in several countries (Italy included), provides a general evaluation of self-perceived psychological health and well-being in the past 4 weeks. The validated Italian short version of the scale [85] comprises 22 polytomous items with scores ranging from 0 to 5 and covers six underlying domains: anxiety (5 items, e.g., “Have you been anxious, worried, or upset during the past month?”), depressed mood (3 items, e.g., “Did you feel depressed during the past month?”), positive well-being (4 items, e.g., “How happy, satisfied, or pleased have you been with your personal life during the past month?”), self-control (3 items, e.g., “I was emotionally stable and sure of myself during the past month”), general health (3 items, e.g., “How often were you bothered by any illness, bodily disorder, aches or pains during the past month?”), and vitality (4 items, e.g., “How much energy, pep, or vitality did you have or feel during the past month?”). In the present study, we used all subscales except for general health which showed low internal consistency in the present study (α = 0.38). Cronbach’s alpha was adequate for all the remaining subscales, ranging from 0.70 to 0.88. Item scores were summed for each subscale and giving them a range of 0 to 100 in order to facilitate comparison across dimensions and studies (see [86]); higher scores indicate greater hedonic psychological well-being. 

The PWB scales [87] investigate eudaimonic psychological well-being and have been validated in Italy [88]. All dimensions of the 42-item version of the scale were used (environmental mastery, self-growth, positive relations with others, purpose in life, and self-acceptance), except for autonomy. Answers were provided on a 6-point Likert scale, ranging from 1 (*definitely disagree*) to 6 (*definitely agree*). Environmental mastery measures the ability to control the surrounding environment and the management of a wide range of activities by taking advantage of opportunities (7 items; e.g., “I am quite good at managing the many responsibilities of my daily life”); self-growth evaluates the sense of continuous growth, expansion, and open-mindedness to experiences and fulfilment of one’s potential (7 items; e.g., “I have the sense that I have developed a lot as a person over time”); positive relations with others assesses trust in people, ability to feel empathy, affection, and ability to create intimate relationships (7 items; e.g., “I know that I can trust my friends, and they know they can trust me”); purpose in life evaluates the presence of a goal and a sense of direction towards life (7 items, one of which was omitted in the present study because it was not consistent with others; e.g., “Some people wander aimlessly through life, but I am not one of them”); self-acceptance assesses positive attitude towards oneself and the awareness of one’s positive and negative qualities (7 items, e.g., “In general, I feel confident and positive about myself”). Cronbach’s alpha was adequate in the present study, ranging from 0.75 to 0.87, for all subscales. A composite score was obtained for each subscale by averaging item scores so that higher scores indicated greater eudaimonic well-being.

*COVID-19 perceived psychological impact.* To evaluate the perceived impact of the pandemic on mental health we used the psychological impact scale of the Coronavirus Impacts Questionnaire–Short Version [89]. The scale consists of 2 highly correlated items (e.g., “The COVID-19 outbreak has impacted my psychological health negatively”; *r* = 0.72).

#### 2.1.3. Data Analysis

All the analyses were carried out using IBM SPSS version 28. 

To test whether the correlations of family resilience with PGWBI and PWB dimensions were significantly different, we first convert each correlation coefficient into a *z*-score using Fisher’s *r*-to-*z* transformation. Then, we used Steiger’s [90] Equations (3) and (10) to compute the asymptotic covariance of the estimates. These quantities were used in an asymptotic *z*-test (see also [91]). 

To test our moderation hypotheses, we used model 1 and model 2 of the SPSS macro PROCESS [92], which allow to estimate moderation models having one and two moderators, respectively. Continuous variables were standardized before entering them into the models to have more comparable coefficients (i.e., partially standardized coefficients).

Finally, to test whether family resilience related to well-being indicators independently of dyadic coping, multivariate regression analyses were performed. To test whether two standardized regression coefficients in the same regression model were significantly different from each other, we used the Wald test [93], whereas to test whether two standardized regression coefficients were statistically significantly different from each other across two different regression models we use the SUEST method, which allows cross-model tests of predictions and marginal effects [94,95].

### 2.2. Results

Descriptive statistics (see Table 2) indicated that on average participants’ levels of family resilience and positive dyadic coping were moderate, and their hedonic and eudaimonic well-being was quite low, especially with reference to the positive well-being, vitality, anxiety, and self-acceptance dimensions. However, the COVID-19 psychological impact was perceived as quite low. All variables were approximately normally distributed, having skewness and kurtosis lower than |2| and |7|, respectively.

Bivariate correlations indicated that family resilience and positive dyadic coping were moderately correlated (*r* = 0.45 ***), confirming that, although associated, they are not overlapping constructs. Also, family resilience was significantly correlated, in the expected direction, with COVID-19 perceived psychological impact (*r* = −0.30 ***), with all PWB dimensions (0.32 *** ≤ *r*s ≤ 0.47 ***) and with depressed mood, positive well-being, and vitality among the PGWBI dimensions (0.19 *** ≤ *r*s ≤ 0.23 ***). When comparing the strength of these correlations through Steiger’s procedure [90], in line with our hypotheses we found that all correlations of family resilience with PWB dimensions were significantly stronger than those with PGWBI dimensions.

PROCESS analyses showed that the associations between family resilience and PGWBI dimensions were moderated by gender (0.32 * ≤ βs ≤ 0.47 ***), employment status [0.42 ** ≤ βs ≤ 0.47 ** when comparing part-time workers (N = 82) to full-time workers (N = 86) or non-working students (N = 130)], and having experienced stressors other than the pandemic since its outbreak (0.23 * ≤ βs ≤ 0.33 **) (see Table 3). Specifically, family resilience was significantly more strongly correlated with PGWBI dimensions in females (0.16 ** ≤ βs ≤ 0.30 ***) than in males (−0.05 ≤ βs ≤ −0.28 *), in part-time workers (0.34 ** ≤ βs ≤ 0.48 ***) than in full-time workers or students who did not have a job (−0.12 ≤ βs ≤ 0.21 *), and in people who experienced other stressors besides the pandemic (0.24 ** ≤ βs ≤ 0.42 ***) than in people who did not (−0.02 ≤ βs ≤ 0.10). Given that employment status was significantly associated with gender (*χ*^2^ (2) = 59.55, *p* = 0.000), with females being less frequently full-time workers and more frequently part-time workers or students compared to males, we tested whether the moderating effects of gender and employment status changed when estimated simultaneously in the same regression model. We found a (marginally) significant unique moderating effect of gender (0.21 ≤ βs ≤ 0.40 *) or employment status (0.13 ≤ βs ≤ 0.37 *) depending on the PGWBI dimension considered, with female part-time workers having the strongest positive associations between family resilience and PGWBI dimensions (0.34 ** ≤ βs ≤ 0.50 ***) (see Appendix A). No other participants’ characteristics among those included in Table 1 were shown to moderate the association between family resilience and well-being indicators (i.e., PGWBI dimensions, PWB dimensions, and COVID-19 psychological impact), not even having a romantic partner or not. This last evidence indicates that the association between family resilience and well-being was similar in the whole sample and in the subsample of people having a romantic relationship.

Multivariate regression analyses carried out on subjects having a romantic partner (*N* = 212) indicated that, even when controlling for positive dyadic coping, family resilience was still significantly related to all well-being indicators except for self-control (0.12 ≤ βs ≤ 0.46 ***) (see Table 4). The SUEST method indicated that the relations of family resilience with PWB dimensions (0.23 ** ≤ βs ≤ 0.46 ***) were in most cases (in 22 out of 25 possible comparisons) stronger than its relations with PGWBI dimensions (0.12 ≤ βs ≤ 0.25 **). Positive dyadic coping had significant associations with positive relations (β = 0.23 **), self-growth (β = 0.20 **), and anxiety (β = −0.20 **) only, when controlling for family resilience. Wald tests indicated that family resilience was positively and uniquely related to all well-being dimensions significantly more strongly than dyadic coping (−0.20 **≤ βs ≤ 20 **), except for its relationship with self-control, self-growth, and positive relations. 

Overall, the results suggest that family resilience was concurrently related to hedonic and eudaimonic well-being as well as with the perceived psychological impact of the COVID-19 pandemic, even though the associations with eudaimonic well-being dimensions were stronger than the ones with hedonic well-being. The same patterns of results were obtained when controlling for dyadic coping in participants having a romantic partner; dyadic coping was less strongly concurrently related to well-being indicators compared to family resilience. Finally, the association between family resilience and hedonic well-being was stronger for some categories of people who had been strongly affected by the pandemic, that is females, part-time workers, and persons who experienced other stressors besides the pandemic.

## 3. Study 2

Study 2 aimed at testing, through a 10-week follow-up, whether: (a) family resilience was differently predictive of hedonic and eudaimonic well-being; (b) the predictive role of family resilience on well-being was moderated by gender, employment status, and exposure to other stressors than COVID-19 pandemic; and (c) for subjects having a romantic partner, family resilience and dyadic coping were significantly predictive of well-being one independently of the other.

On one side, family resilience may be expected to be more predictive of eudaimonic well-being because, as previously discussed, conceptualizations of family resilience and eudaimonic well-being suggest they share many common features [59], which was supported by Study 1 results. On the other side, compared to eudaimonic well-being, hedonic well-being is less stable over time and more strongly affected by emotional experiences which can vary dramatically across time and context [96,97], which probably makes it more susceptible to the beneficial effects of protective factors. The same conflicting assumptions can be made when considering the unique predictive effects of resilience and dyadic coping on the well-being of participants having a romantic partner. Finally, relying on previous cross-sectional evidence, we expected that the protective role of family resilience on well-being might be stronger for females, part-time workers, and people who had been exposed to stressors other than the pandemic since its outbreak.

### 3.1. Methods

#### 3.1.1. Participants and Procedure

One hundred and twelve subjects participated in the follow-up (T2), 80 of whom had a romantic partner. Despite the high attrition rate (48.6% of those who had given their availability to be contacted again), probably due to the concomitant summer holidays after months of lockdowns and restrictive measures, subjects who participated in the follow-up did not significantly differ from those who only participated in the cross-sectional study with respect to any of the characteristics reported in Table 1. Also, a multivariate analysis of variance revealed that participants who provided data for both waves (T1 and T2) were not overall significantly different from those who dropped out after T1 with respect to the variables investigated (*F*_12,312_ = 1.085, *p* = 0.372 for all subjects; *F*_13,198_ = 1.214, *p* = 0.271 for subjects having a romantic partner and also reporting on their dyadic coping).

Participants were contacted through the email address they provided at the end of the previous data collection and invited to complete a new anonymized questionnaire, which took approximately 10 min and was implemented online using the Google Forms platform. Again, no compensation was offered for participation. The new data collection took place in July 2020, 10 weeks after the first data collection, when the pandemic situation had significantly improved and many strict restrictive measures, including the lockdown, were (temporarily) removed. During that period, 2823 Italians were diagnosed with COVID-19 and 184 died after being infected.

Similarly to the previous data collection, all respondents were treated according to the ethical guidelines established by the WMA—Declaration of Helsinki [78] and the Italian Psychological Association [79].

#### 3.1.2. Measures

COVID-19 perceived psychological impact and psychological well-being were assessed through the same measures used in the previous cross-sectional study. All measures proved to have adequate internal consistency (psychological impact scale of the Coronavirus Impacts Questionnaire: *r* = 0.75; PGBWI dimensions: 0.72 ≤ *r*s ≤ 0.88; PWB dimensions: 0.68 ≤ *r*s ≤ 0.87). Composite scores were computed following the same procedures used in Study 1.

Participants were also asked whether they had experienced stressors other than the pandemic after the first data collection, that is in the last 10 weeks; 52.7% of them reported they did (The stressors not caused by the pandemic more frequently reported were: school problems (22.2%), health problems (21.1%), work problems (15.6%), and conflicts with close others (15.6%)). Relying both on this new information and that collected at T1, we computed a categorical variable for stressors experienced, in which 0 = no other stressors had been experienced since the pandemic outbreak (33%), 1 = other stressors had been experience only before T1 or only between T1 and T2 (34.8%), and 2 = other stressors had been experienced both before T1 and between T1 and T2 (32.1%).

#### 3.1.3. Data Analysis

We performed analyses similar to those carried out in Study 1. Specifically, to test whether the correlations among the variables of interests were significantly different, we followed Steiger’s [90] procedure.

We estimated multivariate regression analyses to examine whether family resilience predicted well-being changes over time as well as whether family resilience related to temporal changes in well-being indicators independently of dyadic coping. The Wald test and SUEST method were used to compare regression coefficients in the same regression model and across different regression models, respectively [93,94,95].

Finally, we used model 1 and model 2 of the SPSS macro PROCESS [92] to test our moderation hypotheses, following the same procedure used in Study 1.

### 3.2. Results

All variables were approximately normally distributed, having skewness and kurtosis lower than |2| and |7|, respectively. Compared to T1, on average subjects reported in T2 significantly lower impact of COVID-19 (*t*(111) = 3.815, *p* = 0.000) and higher well-being in terms of anxiety (*t*(111) = −2.131, *p* = 0.035), depression (*t*(111) = −3.084, *p* = 0.003), positive well-being (*t*(111) = −3.115, *p* = 0.002), and environmental mastery (*t*(111) = −2.042, *p* = 0.044) (see Table 5 for descriptives and correlations of variables at T1 with variables at T2). Stability correlation coefficients for PGWBI dimensions across the two waves (0.17 ≤ *r*s ≤ 0.26 **) were significantly lower than stability correlations for PWB (0.68 *** ≤ *r*s ≤ 0.82 ***), when tested through Steiger’s procedure [90].

Both family resiliency and dyadic coping measured at T1 were significantly associated with all well-being indicators assessed at T2, with the exception of self-growth which was unrelated to family resilience, and anxiety and vitality which were unrelated to dyadic coping in subjects having a partner (−0.34 *** ≤ *r*s ≤ 0.54 ***). In 16 cases out of 25, PWB dimensions at T1 were significantly more strongly associated with PGWBI dimensions at T2 (−0.02 ≤ βs ≤ 0.59 ***) than PGWBI dimensions at T1 correlated with PWB dimensions at T2 (−0.10 ≤ βs ≤ 0.18). These results mirror those of Joshanloo [97], which were, however, obtained by investigating hedonic and eudaimonic well-being over much longer periods (i.e., decades).

Multivariate regression analyses indicated that family resilience at T1 (marginally) significantly predicted an increase in all PGWBI dimensions (0.17° ≤ βs ≤ 0.40 ***) and in the PWB positive relations dimension (β = 0.17 **), but not in perceived COVID-19 psychological impact at T2 (β = −0.11), when controlling for their baseline values (see Table 6). The SUEST method indicated that the predictive relations of family resilience with PGWBI dimensions were stronger than its relations with PWB dimensions, except when the PGWBI anxiety dimension was considered.

PROCESS analyses revealed that the predictive effects of family resilience on PGWBI dimensions, when controlling for their baseline values, were mainly moderated by employment status at T1 (0.52 * ≤ βs ≤ 0.76 ** when comparing part-time workers to other groups) and having experienced other stressors since the pandemic outbreak (0.41 * ≤ βs ≤ 0.74 *** when comparing stressors experienced at T1 or T2 with other conditions) (see Table 7). Specifically, family resilience was significantly more predictive of anxiety, depression, and self-control in part-time workers (0.52 * ≤ βs ≤ 0.76 **) than in full-time workers or students who did not have a job (−0.15 ≤ βs ≤ 0.36 **). Also, family resilience was a significantly stronger predictor of PGWBI dimensions in people who experienced other stressors or before T1 or between T1 and T2 (0.35 * ≤ βs ≤ 0.74 ***), compared to people who experienced them in both periods or did not experience them at all (−0.12 ≤ βs ≤ 0.41 *). Contrary to prediction, we found that family resilience was significantly more predictive of positive well-being for males (β = 0.81 **) than for females (β = 0.31 **; interaction effect: β = −0.50 *). Given that employment status was significantly associated with gender (*χ*^2^ (2) = 21.81, *p* = 0.000) even for subjects participating at the follow-up, we verified whether the moderating effects of gender and employment status changed when estimated simultaneously in the same regression model through PROCESS model 2, as we did in Study 1. We found a significant unique moderating effect of gender and employment status for depressed mood (β = −0.69 ** and 1.06 **, respectively) and positive well-being (β = −0.74 ** and 0.63 *, respectively), with male part-time workers having the strongest associations between family resilience and these PGWBI dimensions (β = 1.34 ** and 1.32 **), even though the conditional effects for female part-time workers were significant as well (β = 0.65 *** and 0.59 ***) (see Appendix A). The predictive effects of family resilience on PWB dimensions and perceived COVID-19 psychological impact were not moderated by gender, employment status, or having experienced stressors other than the pandemic. As before, having a romantic partner or not was not a significant moderator of the longitudinal links between family resilience and well-being, suggesting that family resilience was predictive of well-being similarly in the overall follow-up sample and its subsample of people having a romantic relationship.

Multivariate regression analyses carried out on subjects having a romantic partner (*N* = 80) indicated that, even when controlling for dyadic coping at T1, the family resilience (marginally) significantly predicted an increase in PGWBI positive well-being, self-control, and vitality dimensions (0.23° ≤ βs ≤ 0.26 *), but not in PWB dimensions, nor was family resilience longitudinally related to perceived COVID-19 psychological impact (see Table 8). The SUEST method showed that predictive relationships of family resilience with PGWBI positive well-being and self-control dimensions (βs = 0.24 * and 0.26 *, respectively) were significantly stronger than its relations with PWB dimensions (βs= 0.11 and −0.08, respectively). When controlling for family resilience, positive dyadic coping had significant relations with all PGWBI dimensions, except for vitality, and with PWB self-growth, positive relations, and self-acceptance dimensions (0.07 ≤ βs ≤ 0.36 **). Wald tests indicated that positive dyadic coping was more strongly uniquely related to anxiety and self-growth (βs = 0.26 * and 0.15, respectively) than family resilience (βs = −0.02 and 0.02, respectively).

In summary, with regard to the variation in the well-being indicators over time, eudaimonic well-being was substantially stable over time, with the exception of environmental mastery which was higher during the follow-up. On the contrary hedonic well-being improved over a 10-week period, especially in terms of increased positive well-being and reduced anxiety, depression, and perceived psychological impact of COVID-19. Also, eudaimonic well-being was more predictive of hedonic well-being than vice versa.

Family resilience was more predictive of hedonic than eudaimonic well-being, even when controlling for positive dyadic coping among participants having a romantic partner. Family resilience was more strongly longitudinally related to hedonic well-being among part-time workers, especially if males, and people who experience other stressors than the pandemic before the first data collection or between the first and the second data collection, that is, during a period of 2.5 months at the most.

Finally, when controlling for family resilience, positive dyadic coping was predictive of greater hedonic and eudaimonic well-being among people having a romantic partner; also, dyadic coping was longitudinally related with some indicators of well-being to a significantly stronger extent than family resilience.

## 4. Discussion

The results indicate that family resilience was concurrently related to both hedonic and eudaimonic well-being, but more strongly related to the latter than to the former, consistent with the theorization of family resilience and eudaimonia, which are assumed to entail similar features like positive personal transformation and growth, a perception of life as meaningful, and strengthened social relations [59]. Our results are also in line with existing studies [48,49,50] documenting that family resilience was concurrently associated with hedonic well-being during the COVID-19 pandemic but extend them by showing that family resilience is also associated with eudaimonic well-being.

Despite the strong concurrent correlations, family resilience was less predictive of changes in eudaimonic than hedonic well-being over a relatively brief period (10 weeks). This finding might be attributable to the much lower variability observed in the levels of eudaimonic than hedonic well-being. Consistently, previous studies documented that, being more affected by emotional experiences which can vary radically across time and context, hedonic well-being is more variable over time, compared to eudaimonic well-being [96,97]. To the best of our knowledge, this is among the first research which shows the predictive role of family resilience on well-being experienced during the COVID-19 pandemic. The longitudinal research on the topic is not only very limited, but even inconsistent in its findings: one study found that family resilience mitigates the negative impact of the pandemic on adults’ depression and anxiety [56], whereas another study, examining the same variables in a small sample of adolescents, found no significant relations among them over time [98]. The present research suggests that family resilience can actually have a protective role in the short term especially on hedonic well-being, thereby extending previous research documenting the cross-sectional association of family resilience with negative affect, depression, anxiety, and stress experienced during pandemic period [51,52,53,54,56,98].

Differently from studies previously conducted during the COVID-19 pandemic, our research also explored whether the links between family resilience and well-being dimensions were moderated by a wide range of socio-demographic and pandemic-related characteristics. The concurrent and longitudinal links between family resilience and hedonic well-being were moderated by three of the many participants’ characteristics considered: gender, employment status, and stressors undergone. Indeed, such links were mostly stronger for women, part-time workers, and persons having faced with other stressful events simultaneously with the pandemic, that is, for categories of people particularly at high risk of low mental health during the COVID-19 pandemic [99,100]. Specifically, family resilience was more strongly related to hedonic well-being among women than men concurrently but not longitudinally when we found a stronger predictive role of resilience for men’s positive well-being than women’s. Probably, since women were more negatively affected by the pandemic than men (e.g., [15,101]), they immediately benefitted from a more resilient and supportive family climate, whereas men experienced a sort of “delayed effect”. This could be explained by the fact that men, having a more independent self-construal than women [102], may prefer to first cope with challenging situations by relying on their own resources and turn to family ones only later, in case of failure of individual efforts. Under conditions of workday stress, for example, men in heterosexual couples are more likely than women to withdraw from marital interactions (e.g., [103]), thereby suggesting that men tend to deal more with their stress arousal by themselves than women. Moreover, family resilience was more strongly (concurrently and longitudinally) related to hedonic well-being among part-time workers than full-time workers or students. Compared to the latter, the former were more exposed to job and income losses and likely to suffer economic distress during and after the first COVID-19 lockdowns (e.g., [104,105]); greater job insecurity and financial concerns during the pandemic were related to higher anxiety and depression symptoms [106,107,108]. Therefore, being more vulnerable to the negative consequences of the pandemic, part-time workers, like women, could have benefitted more from the protective role of family resilience. Even people who had faced other stressful events in addition to the pandemic showed stronger relations between family resilience and hedonic well-being. According to the adaptive cost model of stress [109,110] people facing multiple stressors are more likely to have poorer mental health due to fatigue and depleted resources resulting from coping efforts (e.g., [111,112]). Our results indicated that high levels of family resilience were able to protect from these undesirable mental health outcomes but only in case exposure to multiple stressors during the pandemic occurred in a limited time period, namely in less than three months. If the presence of multiple stressors lasts longer, family resilience was insufficient to mitigate their negative impact on hedonic well-being. Overall, these moderating effects are of primary practical importance, as they allowed identifying those groups of population—namely part-time workers and persons exposed to multiple stressors in a limited time period—for whom family resilience interventions could be more effective in protecting from COVID-19 undesirable mental health outcomes.

Despite the close interconnections between family resilience and dyadic coping [41,49,71,72,73], this research is the first to investigate their unique contributions to psychological well-being in people having a partner. Specifically, when considering people involved in a romantic relationship, family resilience was concurrently associated with almost all well-being dimensions and longitudinally related only to a few hedonic well-being dimensions, independently of positive dyadic coping. On the contrary, when controlling for family resilience, positive dyadic coping was concurrently related to a few hedonic and eudaimonic well-being dimensions, but it was predictive of most well-being dimensions over time. Thus, despite family resilience and dyadic coping being supposed to be related and having similar outcomes [41,49], their unique contribution to psychological well-being seems to differ. In particular, family resilience was concurrently associated with both hedonic and eudaimonic well-being, but it was not strongly predictive of them in the short period. This suggests that, when people face highly challenging and stressful events, such as a pandemic, the resilience capacities of their family may take a long time to progressively adapt to the disturbances caused by adversities and manifest their positive effects. In line with our results, Conger and Conger [70] found that resilience skills exhibited during major life stressors promote psychological well-being over decades. Consistently, Walsh’s [41] developmental perspective of family resilience assumes that it typically requires a long time to emerge and display its effects (see also [66]). Conversely, our findings suggest that positive dyadic coping may have more immediate effects on psychological well-being, consistent with previous research showing that it can be associated with outcomes even on a daily basis [67,68]. It is also possible that, for people having a romantic partner, addressing pandemic challenges with the partner is more relevant to their psychological well-being than being able to do it with the whole family, especially in a Western, individualistic context (e.g., [76,77]).

With regards to practical implications, our results suggest that interventions fostering family resilience among people exposed to consistent community and individual distressing situations, like those caused by the COVID-19 pandemic outbreak, may have positive health outcomes in the short term. As suggested by Walsh’s family resilience framework as well as existing evidence on community stressors [48,113], intervention programs should be tailored to strengthen positive perceptions, cohesion, adaptability, communication, coping strategies, and adequate financial management within families. Informed by this framework, Ruiz and colleagues have for example quickly developed, in the aftermath of the COVID-19 pandemic outbreak, the *Families Tackling Tough Times Together* (FT) program, specifically designed to enhance family resilience during the pandemic. The program consists of short videos with self-guided family activities which had been disseminated weekly through a public Facebook group and a website for 9-11 weeks. Unfortunately, the authors have been unable to empirically support the program effectiveness, probably because they tested it through measures of trait-like constructs, which were hard to change in the short period. Our results indicate that hedonic well-being dimensions are likely to be more appropriate variables to be assessed when validating family resilience interventions.

Our findings also suggest that those who may benefit most from such interventions are people most at risk of mental health impairments in community challenging situations, like part-time workers who suffer from greater job and economic insecurity, and persons dealing with multiple stressful events. Consistent with our results, the Employment Precarity Family Stress model claims that family coping and resilience are crucial to adapt to employment precarity and the multiple stressors it implies. However, we must not forget that family coping and resilience are strongly interdependent with the wider socio-economic context in which the family is embedded. Therefore, to promote family resilience in the face of extremely challenging situations, such as the COVID-19 pandemic has been, it is essential not only to strengthen family functioning through clinical work, but also to address the structural problems of the larger community and societal institutions (e.g., employment precarity, inadequate health care, insufficient childcare and disability services) which prevent families from having control over their lives and facing adversities in an adaptive manner [41].

For people involved in a romantic relationship, it may also be particularly important to enhance dyadic coping skills through interventions such as the *Couples Coping Enhancement Training* (CCET; [114]) and, especially in the case of financial stress, the Together program [115]. In particular, our findings on the role of dyadic coping for well-being during emergency situations emphasize the need for a dyadic, rather than individual, approach to interventions aimed at sustaining partners’ mental health during stressful circumstances. Stress-reduction programs, when stress is experienced in the context of a couple relationship, should in fact tackle not only the individuals’ stress appraisals and coping skills, but also the couple’s dyadic coping and the interdependence of both partners’ stress levels, distress symptoms, and coping strategies.

More generally, our research indicates that both family resilience and dyadic coping should be evaluated as potential targets of public health policies, because of their ability to counter some negative psychological consequences of the pandemic, especially among the most vulnerable population, like part-time workers and persons dealing with multiple stressful events.

### Strengths, Limitations, and Future Directions

The research has a number of strengths. To the best of our knowledge, it is the only research which examined the concurrent and longitudinal associations of family resilience with not only hedonic but also eudaimonic well-being experienced during the pandemic as well as tested whether such associations were moderated by a wide range of socio-demographic and pandemic-related characteristics. Also, our research investigated whether family resilience and dyadic coping—two constructs assumed to be closely interrelated but rarely analyzed simultaneously—were associated with and predictive of psychological well-being, one independently of the other. Finally, the research explores these questions a few weeks after the COVID-19 pandemic outbreak in Europe right at its epicenter, therefore on people strongly affected by it.

As with all studies, however, the present one has some limitations. The sample was one of convenience; participants were selected according to their availability at the moment of data collection and their willingness to take part in the study. This led to a gender-biased sample with a high follow-up attrition, which may have negatively impacted the generalizability and the statistical power of the findings. Indeed, it should be emphasized that our follow-up study involved a very limited number of subjects (having or not a partner), therefore conclusions derived from it must be interpreted with caution. The limited number of participants in the follow-up study also prevented us from testing the longitudinal links between the variables of interest using Structural Equation Modeling (SEM), which has several advantages over regression analyses (e.g., modeling measurement errors and unexplained variances, simultaneous testing of multiple paths) but needs larger samples to have adequate statistical power. Future research should test our results through SEM on a larger and more heterogeneous group of individuals with respect to gender, age, romantic involvement, and marital status, so as to provide a more reliable and comprehensive understanding of the relations of family resilience and dyadic coping with psychological well-being.

Moreover, even though the longitudinal design contributed to clarifying the directionality of the links from family resilience or dyadic coping to well-being, it was not possible to explore the opposite longitudinal links from well-being to family resilience or dyadic coping since these last two variables had been measured in the first wave only. Consequently, further longitudinal studies are needed to fully understand the reciprocal relationships between these constructs over time. Possibly, such studies should be conducted over longer time periods, so as to examine whether the positive effects of family resilience are more pronounced and extend to the eudaimonic dimensions of well-being in the long term, as predicted by family resilience theories [41,66].

Finally, studies are needed to test whether the protective role of family resilience and dyadic coping on psychological well-being may differ depending not only on socio-demographic and COVID-19 related characteristics, but also on the family members’ and community’s resources and abilities. According to the ecosystemic view of the family [116], family resilience is indeed affected by the interplay of many individual, family, community, and larger system variables.

## 5. Conclusions

In summary, our findings indicate that family resilience protects, in the short term, the psychological well-being of people who face very challenging situations, like the COVID-19 outbreak. Its protective role mainly concerns the hedonic components of well-being and is more pronounced for more vulnerable people, like part-time workers and persons undergoing multiple stressors. For persons having a romantic partner, however, dyadic coping seems to have equal, if not greater, positive short-term effects.

Our findings have significant implications for designing prevention and intervention programs aimed at protecting well-being during extremely challenging times. They suggest that both prevention and intervention should focus on family and dyadic dynamics in addition to individual functioning. Specifically, fostering family resilience and dyadic coping among people exposed to community stressors looks promising, in particular for people undergoing economic strains and suffering multiple stressors. Past studies indicated that family resilience and dyadic coping can be sustained by systematic interventions [41,114,115]; however, more studies are needed to examine whether such interventions are effective even in a pandemic situation.

## Figures and Tables

**Table 1 ijerph-20-06719-t001:** Participants’ characteristics.

Characteristic	
Age	*M* = 29.9; *SD* = 12.8
Sex (%)	
Female	76.0
Male	24.0
Degree (%)	
High-school	50.8
Bachelor	24.6
Master	13.5
Others	11.1
Employment status (%)	
Full-time	27.0
Part-time	25.8
Students not working	40.9
Others	6.3
Workers who had lost job due to pandemic (%)	
No	55.4
Yes, subsidized lay-off	26.5
Yes, lay-off	18.1
COVID-19 risk at work	*M* = 5.5; *SD* = 3.2 ^A^
Had a romantic partner (%)	
No	34.8
Yes, a non-cohabiting romantic partner	40.0
Yes, a cohabiting romantic partner	25.2
Marital status (%)	
Not married	79.4
Married	18.2
Separated or divorced	2.2
Widow(er)	0.3
Lived with parents (%)	
No	35.4
Yes	64.6
Lived with different people than usual due to pandemic (%)	
No	83.4
Yes	16.6
House m^2^ per person	*M* = 42.8; *SD* = 29.7
Garden m^2^ per person	*M* = 84.9; *SD* = 357.6
Political orientation	*M* = 4.5; *SD* = 4.5 ^B^
Had experienced distressing events not caused by the pandemic since its outbreak	
No	58.50%
Yes	41.50% ^C^

Note: ^A^ 1 = definitely low risk; 10 = definitely high risk; ^B^ 1 = left; 9 = right. *N* = 325; ^C^ The stressors not caused by the pandemic more frequently reported were: conflicts with close others (17.4%), school problems (12.3%), work problems (12.3%), health problems (11.4%), and distance from loved ones (10.4%).

**Table 2 ijerph-20-06719-t002:** Descriptive statistics and Pearson’s correlations among variables—Study 1.

	*M*	*SD*	Range	Skewness	Kurtosis	1	2	3	4	5	6	7	8	9	10	11	12
1. FAMILY RESILIENCE	3.4	0.6	1.5–5	−0.15	0.58												
2. DYADIC COPING	3.9	0.5	2.2–5	−0.55	0.45	0.45 ***											
PGWBI																	
3. Anxiety	59.4	19.8	0–100	−0.33	−0.12	0.11	−0.13										
4. Depressed mood	77.3	16.2	7–100	−1.69	4.00	0.23 ***	−0.03	0.69 ***									
5. Positive well-being	47.1	16.4	5–100	0.45	0.37	0.21 ***	−0.01	0.63 ***	0.70 ***								
6. Self-control	67.1	19.6	0–100	−0.34	−0.38	0.09	−0.07	0.70 ***	0.68 ***	0.66 ***							
7. Vitality	57.5	17.4	5–100	−0.29	−0.26	0.19 ***	−0.02	0.68 ***	0.73 ***	0.78 ***	0.69 ***						
PWB																	
8. Environmental mastery	4.1	0.9	1.3–6	−0.50	0.03	0.46 ***	0.23 **	0.19 **	0.25 ***	0.21 ***	0.22 ***	0.24 ***					
9. Self-growth	4.7	0.7	2.1–6	−0.32	−0.08	0.32 ***	0.30 ***	0.09	0.16 **	0.12 *	0.10	0.11	0.50 ***				
10. Positive relations	4.6	0.8	1.7–6	−0.81	0.38	0.46 ***	0.38 ***	0.07	0.17 **	0.18 **	0.07	0.19 **	0.59 ***	0.47 ***			
11. Purpose in life	4.3	0.9	1.7–6	−0.46	−0.28	0.46 ***	0.18 *	0.15 **	0.25 ***	0.22 **	16 **	0.24 ***	0.70 ***	0.49 ***	0.53 ***		
12. Self-acceptance	4.0	1.1	1.1–6	−0.31	−0.39	0.47 ***	0.24 **	0.19 **	0.27 ***	0.26 ***	0.22 ***	0.25 ***	0.76 ***	0.49 ***	0.62 ***	0.64 ***	
13. CPI	2.8	1.6	1–7	0.78	−0.37	−0.30 ***	−0.12	−0.23 ***	−0.27 ***	−0.23 ***	−0.23 ***	−0.24 ***	−0.39 ***	−0.14 *	−0.19 **	−0.19 **	−0.35 ***

CPI = COVID-19 Psychological Impact; PGWBI = Personal General Well-Being Index; PWB = Personal Well-Being scale. *N* = 325, except for dyadic coping and its correlations (*N* = 212). * *p* < 0.05; ** *p* < 0.01; *** *p* < 0.001.

**Table 3 ijerph-20-06719-t003:** Partially standardized regression coefficients and percentage of variance explained for sample characteristic which significantly moderated the association between family resilience and PGWBI dimensions—Study 1.

Moderator		Anxiety	Depressed Mood	Positive Well-Being	Self-Control	Vitality
Gender	Males	−0.28 *	−0.05	−0.05	−0.17	−0.060
	Females	0.19 **	0.30 ***	0.27 ***	0.16 **	0.26 ***
	*Interaction coefficient*	*0.47* ***	*0.36* *	*0.32* *	*0.33* *	*0.32* *
	Δ*R*^2^ *interaction effect*	*0.04* **	*0.02* *	*0.02* *	*0.02* *	*0.02* *
Employment status	Full-time workers	−0.11	0.06		−0.12	−0.01
	Part-time workers	0.34 **	0.48 ***		35 **	0.41 ***
	Not working students	0.12	0.19 *		0.08	21 *
	*Interaction coefficient for part-time workers vs. full-time workers or students*	*0.46* ***	*0.42* **		*0.47 ***	*0.42* **
	*Interaction coefficient for students vs. part-time or full-time workers*	*0.23*	*0.13*		*0.20*	*0.22*
	Δ*R*^2^ *interaction effect*	*0.03* *	0.02 *		*0.03* *	*0.02* *
Experienced distressing events other than pandemic since its outbreak	No	−0.02	0.10		−0.01	0.10
	Yes	0.31 ***	0.42 ***		0.24 **	0.32 ***
	*Interaction coefficient*	*0.33* **	*0.32* **		*0.25* *	*0.23* *
	Δ*R*^2^ *interaction effect*	*0.03* **	*0.02* *		*0.02* *	*0.01* *

PGWBI = Personal General Well-Being Index. *N* = 325, except for employment status moderation (N = 298). * *p* < 0.05; ** *p* < 0.01; *** *p* < 0.001. Interaction coefficients and percentage of variance explained by them are reported in italics.

**Table 4 ijerph-20-06719-t004:** Standardized regression coefficients and percentage of variance explained for family resilience and dyadic coping predicting well-being indicators—Study 1.

	PGWBI					PWB					CPI
Predictors	Anxiety	Depressed Mood	Positive Well-Being	Self-Control	Vitality	Environmental Mastery	Self-Growth	Positive Relations	Purpose in Life	Self-Acceptance	
Family resilience	0.16 *	0.25 **	0.24 **	0.12	0.22 **	0.43 ***	0.23 **	0.33 ***	0.46 ***	0.43 ***	−0.29 ***
Positive dyadic coping	−0.20 **	−0.14	−0.12	−0.12	−0.12	0.04	0.20 **	0.23 **	−0.03	0.04	0.00
*R* ^2^	0.04 *	0.05 **	0.05 **	0.02	0.04 *	0.20 ***	0.13 ***	0.23 ***	0.20 ***	0.21 ***	0.08 ***

CPI = COVID-19 Psychological Impact; PGWBI = Personal General Well-Being Index; PWB = Personal Well-Being scale. *N* = 212. * *p* < 0.05; ** *p* < 0.01; *** *p* < 0.001.

**Table 5 ijerph-20-06719-t005:** Descriptive statistics and Pearson’s correlations of variables at T1 with variables at T2—Study 2.

						Variables Measured at T2										
	*M*	*SD*	Range	Skew-ness	Kurt-osis	3	4	5	6	7	8	9	10	11	12	13
*M*						62.9	81.4	53.2	68.5	57.5	4.2	4.7	4.6	4.3	4.0	2.6
*SD*						18.6	12.0	17.3	18.3	17.1	0.9	0.7	0.9	0.9	1.0	1.5
Range						16–96	40–100	20–90	27–100	20–95	1.9–5.9	1.7–5.9	1.7–5.9	1.8–6	1–6	1–7
Skewness						−0.56	−1.10	0.10	−0.50	−0.16	−0.65	−0.98	−1.21	−0.41	−0.39	1.07
Kurtosis						−0.39	1.16	−0.87	−0.52	−0.61	−0.20	2.12	1.49	−0.38	−0.29	0.38
Variables measured at T1																
1. FAMILY RESILIENCE	3.3	0.6	1.8–4.6	−0.39	−0.09	**0.19 ***	**0.39 *****	**0.42 *****	**0.40 *****	**0.33 *****	**0.48 *****	**0.14**	**0.54 *****	**0.34 *****	**0.49 *****	**−0.34 *****
2. DYADIC COPING	3.9	0.5	2.45–4.8	−0.69	0.53	**0.21**	**0.42 *****	**0.35 ****	**0.37 ****	**0.15**	**0.32 ****	**0.32 ****	**0.39 ****	**0.31 ****	**0.33 ****	**−0.30 ****
PGWBI																
3. Anxiety	58.2	19.6	0–100	−0.36	0.08	**0.26 ****	0.13	0.18	0.10	0.19 *	0.09	−0.10	−0.03	0.07	0.18	−0.11
4. Depressed mood	76.0	16.7	7–100	−1.78	4.64	0.14	**0.19 ***	0.17	0.09	0.17	0.13	0.00	−0.04	0.12	0.11	−0.16
5. Positive well-being	46.9	16.5	5–100	0.30	0.63	0.15	0.17	**0.19 ***	0.13	0.20 *	0.11	0.04	0.02	0.06	0.12	−0.19 *
6. Self-control	66.9	19.7	0–100	−0.55	0.29	0.20 *	0.23 *	0.22 *	**0.17**	0.11	0.08	−0.07	−0.10	0.11	0.17	−0.15
7. Vitality	56.6	17.8	15–100	−0.22	−0.34	0.18	0.22 *	0.25 **	0.16	**0.25 ****	0.16	0.01	0.05	0.11	0.16	−0.15
PWB																
8. Environmental mastery	4.1	0.9	1.3–6	−0.64	0.37	0.31 **	0.53 ***	0.52 ***	0.59 ***	0.44 ***	**0.72 *****	0.32 **	0.57 ***	0.58 ***	0.66 ***	−0.46 ***
9. Self-growth	4.6	0.7	2.1–5.9	−0.72	0.91	−0.02	0.17	0.21 *	0.26 **	0.15	0.40 ***	**0.73 *****	0.43 ***	0.50 ***	0.42 ***	−0.18
10. Positive relations	4.6	0.8	2.14–5.9	−0.90	0.63	0.17	0.34 ***	0.36 ***	0.40 ***	0.28 **	0.53 ***	0.44 ***	**0.82 *****	0.43 ***	0.49 ***	−0.25 **
11. Purpose in life	4.3	0.9	2–5.8	−0.42	−0.49	0.13	0.32 **	0.42 ***	0.46 ***	0.31 **	0.53 ***	0.42 ***	0.53 ***	**0.76 *****	0.59 ***	−0.23 *
12. Self-acceptance	3.9	1.0	1.3–5.9	−0.29	−0.44	0.34 ***	0.46 ***	0.54 ***	0.58 ***	0.48 ***	0.65 ***	0.33 ***	0.52 ***	0.59 ***	**0.81 *****	−0.39 ***
13. CPI	3.0	1.6	1–7	0.49	−0.80	−0.29 **	−0.39 ***	−0.37 ***	−0.41 ***	−0.30 **	−0.29 **	−0.06	−0.08	−0.20 *	−0.37 ***	**0.68 *****

CPI = COVID-19 Psychological Impact; PGWBI = Personal General Well-Being Index; PWB = Personal Well-Being scale. *N* = 112 except for dyadic coping and its correlations (*N* = 80). Descriptives for variables at T1 were reported in the first 5 columns, whereas descriptives for variables at T2 were reported in the first 5 lines. Stability correlation coefficients as well as coefficients for correlation between family resilience and dyadic coping at T1 and well-being indicators at T2 are reported in bold characters. * *p* < 0.05; ** *p* < 0.01; *** *p* < 0.001.

**Table 6 ijerph-20-06719-t006:** Standardized regression coefficients and percentage of variance explained for family resilience at T1 predicting well-being indicators at T2, when controlling their baseline values—Study 2.

	PGWBI					PWB					CPI
Predictors	Anxiety	Depressed Mood	Positive Well-Being	Self-Control	Vitality	Environmental Mastery	Self- Growth	Positive Relations	Purpose in Life	Self-Acceptance	
Family resilience	0.17 °	0.37 ***	0.40 ***	0.39 ***	0.29 *	0.11	−0.06	0.17 **	−0.02	0.06	−0.11
Well-being indicator at T1	0.26 **	14	0.11	0.13	0.20 **	0.67 ***	0.74 ***	0.74 ***	0.77 ***	0.77 ***	0.64 ***
*R^2^*	0.10 **	0.17 ***	0.19 ***	0.18 ***	0.15 ***	0.53 ***	0.53 ***	0.70 ***	0.58 ***	0.65 ***	0.47 ***

CPI = COVID-19 Psychological Impact; PGWBI = Personal General Well-Being Index; PWB = Personal Well-Being scale. *N* = 112. ° *p* < 0.06; * *p* < 0.05; ** *p* < 0.01; *** *p* < 0.001.

**Table 7 ijerph-20-06719-t007:** Partially standardized regression coefficients and percentage of variance explained for sample characteristic which significantly moderated the association between family resilience at T1 and PGWBI dimensions at T2, when controlling for their baseline values—Study 2.

Moderator		Anxiety	Depressed Mood	Positive Well-Being	Self-Control	Vitality
Gender	Males			0.81 ***		
	Females			0.31 **		
	*Interaction coefficient*			*−0.50* *		
	Δ*R*^2^ *interaction effect*			*0.03* *		
Employment status at T1	Full-time workers	−0.15	−0.07		0.18	
	Part-time workers	0.44 *	0.69 ***		0.56 **	
	Not working students	0.14	0.36 **		0.36 **	
	*Interaction coefficient for part-time workers vs. full-time workers or students*	*0.59* *	*0.76* **		*0.52* *	
	*Interaction coefficient for students vs. part-time or full-time workers*	*0.29*	*0.43* °		*0.32*	
	Δ*R*^2^ *interaction effect*	0.04	0.07 *		0.03	
Experienced distressing events other than pandemic since its outbreak	No	−0.12	0.00	0.14	0.18	0.07
	At T1 or T2	0.35 *	0.74 ***	0.62 ***	0.59 ***	0.49 **
	At T1 and T2	0.33 °	0.32 °	0.41 *	0.37 *	0.35 °
	*Interaction coefficient for at* T1 *or* T2 *vs. others*	*0.47* *	*0.74* ***	*0.47* *	*0.41* *	*0.41* *
	*Interaction coefficients for at* T1 *and* T2 *vs. others*	*0.44* °	*0.32*	*0.27*	*0.20*	*0.28*
	Δ*R*^2^ *interaction effect*	*0.05* *	*0.10* ***	*0.04* °	*0.03*	*0.03*

PGWBI = Personal General Well-Being Index. *N* = 112 except for employment status moderation (*N* = 101). ° *p* < 0.06; * *p* < 0.05; ** *p* < 0.01; *** *p* < 0.001. Interaction coefficients and percentage of variance explained by them are reported in italics.

**Table 8 ijerph-20-06719-t008:** Standardized regression coefficients and percentage of variance explained for family resilience and dyadic coping at T1 predicting well-being indicators at T2, when controlling their baseline values—Study 2.

	PGWBI					PWB					CPI
Predictors	Anxiety	Depressed Mood	Positive Well-Being	Self-Control	Vitality	Environmental Mastery	Self- Growth	Positive Relations	Purpose in Life	Self-Acceptance	
Family resilience at T1	−0.02	0.20	0.24 *	0.26 *	0.23 °		0.02	−0.14	0.11	−0.08	0.01
Dyadic coping at T1	0.26 *	0.36 **	0.27 *	0.27 *	0.07		0.15	0.18 *	0.16 *	0.13	0.15 *
Well-being indicator at T1	0.38 **	0.21 ***	0.11	0.09	0.12		0.64 ***	0.69 ***	0.73 ***	0.79 ***	0.73 ***
*R* ^2^	0.19 **	0.17 ***	0.15 **	0.21 ***	0.10 °		0.50 ***	0.52 ***	0.72 ***	0.62 ***	0.62 ***

CPI = COVID-19 Psychological Impact; PGWBI = Personal General Well-Being Index; PWB = Personal Well-Being scale. *N* = 80. ° *p* < 0.06; * *p* < 0.05; ** *p* < 0.01; *** *p* < 0.001.

## Data Availability

Data reports and datasets can be obtained by emailing the lead author and coordinating a data sharing agreement: Francesca Giorgia Paleari (francesca-giorgia.paleari@unibg.it).

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
