# Peer review of "Family Resilience and Dyadic Coping during the Outbreak of the COVID-19 Pandemic in Italy: Their Protective Role in Hedonic and Eudaimonic Well-Being"

_ijerph, 2023, doi:10.3390/ijerph20186719_

Round 1

Reviewer 1 Report

Family resilience and dyadic coping during the outbreak of  COVID-19 pandemic in Italy:Their protective role on hedonic  and eudaimonic well-being

1.     It is recommended to provide explicit definitions at the outset of the article to ensure clarity and eliminate ambiguity associated with potentially equivocal terms. Several terms and concepts may benefit from clearer definitions or explanations. For instance, providing a concise definition of hedonic well-being and eudaimonic well-being would help readers better understand the specific aspects of well-being being examined. Same with the words resilience and dyadic coping.

2.     The article could benefit from a discussion of the theoretical and practical implications of the findings. How do the results align with or contribute to existing theories or frameworks in the field of resilience and well-being? Highlighting the practical relevance of the results, such as implications for interventions or support programs targeting specific groups, would make the findings more actionable and relevant for practitioners and policymakers. Addressing these implications would enhance the findings' theoretical significance and offer a more nuanced understanding of the mechanisms and processes involved.

3.     I observed that using a convenience sample introduced bias. The sample's gender distribution was imbalanced, and there was a high attrition rate during the follow-up, potentially impacting the statistical power and the generalizability of the findings. Including a larger and more diverse group of individuals, including those without romantic partners, could have provided a more comprehensive understanding of the impact of family resilience and dyadic coping on well-being.

4.     For your recommendation, I suggest that since the study was unable to explore the reverse longitudinal links from well-being to family resilience or dyadic coping due to their measurement only in the initial wave, there may be a need for future study to fully understand the reciprocal relationships between these constructs over time.

5.     The study is able to contribute to the complex interplay between family resilience, dyadic coping, and well-being in the context of crises like the COVID-19 pandemic.

Moderate English editing may be needed. 

Reviewer 2 Report

Thank you for the opportunity to read this  interesting paper. The paper is very interesting, but I think some minor revisions are needed. The following suggestions could improve the the presentation for publication.

1) Statistical analysis performed can be included in the abstract. 

2)Concerning the paragraph  "Research overview", I think it is unclear: both its function and  the hypotheses (as they are formulated) are unclear. 

3) it's nedeed included in the descriptive analysis section skewness and kurtosis

4) Regarding the results, i think you need to improve the presentation. you can included   the B and p value for the moderation results. Likewise, statistical results should be included for the regression analysis. 

5) think it is needed to specify in the table 5 the variables measured at time 1 and time 2. 

6)Why didn't you test the longitudinal hypotheses with structural equation models?

7)There is only a short Conclusion. There is a lack of practical implications and directive for future studies
